

# Production and purification of mannan oligosaccharide with epithelial tight junction enhancing activity

Chatchai Nopvichai[1], Thanapon Charoenwongpaiboon[1], Navaporn Luengluepunya[1], Kazuo Ito[2], Chatchai Muanprasat[3] and Rath Pichyangkura[1]

[1] Department of Biochemistry, Faculty of Science, Chulalongkorn University, Bangkok, Thailand
[2] Graduate School of Science, Osaka City University, Osaka, Japan
[3] Department of Physiology, Faculty of Science, Mahidol University, Bangkok, Thailand

## ABSTRACT

**Background.** Mannanan oligosaccharide (MOS) is well-known as effective supplement food for livestock to increase their nutrients absorption and health status, but the structure and identification of bioactive MOS remain unclear. In this study, MOS production was accomplished, using enzymatic hydrolysis of pretreated coconut meal substrate with recombinant mannanase.

**Methods.** The mannanase gene was cloned from *Bacillus subtilis c*AE24, then expressed in BL21. Purified Mannanase exhibit stability over a wide range of pH and temperature from pH 6–8 and 4 °C to 70 °C, respectively. SEM analysis revealed that sonication could change the surface characteristic of copra meal, which gave better MOS yield, compared to untreated substrates. The separation and purification of each MOS were achieved using Biogel-P2 column chromatography. Determination of biological active MOS species was also investigated. T84 cells were cultured and treated with each of the purified MOS species to determine their tight junction enhancing activity.

**Results.** Scanning electron microscope imaging showed that pretreatment using sonication could disrupt the surface of copra meal better than grinding alone, which can improve the production of MOS. Pentamer of MOS (M5) significantly increased tight junction integration of T84 cells measured with TEER ($p < 0.0001$).

## INTRODUCTION

Mannan oligosaccharide (MOS) is an indigestible short chain polymer and a well-known supplement for increasing the life quality of pets and livestock. Several reports revealed that MOS supplement could improve growth performance and body weight in various animals (*Ai et al., 2011*; *Dimitroglou et al., 2010*; *Genc et al., 2007*; *Mansour et al., 2012*; *Staykov et al., 2007*). It can enhance the immunity and also metabolic and stress response of aquamarine cultures including shrimp, seabream, sturgeon and rainbow trout (*Ai et al., 2011*; *Dimitroglou et al., 2010*; *Mansour et al., 2012*; *Ozaki, Fujii & Hayashi, 2007*; *Rungrassamee et al., 2014*; *Staykov et al., 2007*). Besides, the beneficial effects of MOS are observed in terrestrial animals like poultry and mammals as it raises nutrient digestibility,

Corresponding author
Rath Pichyangkura, prath@chula.ac.th

cecal fermentation, and improves the intestinal morphology (*Cheled-Shoval et al., 2011*; *Cheled-Shoval et al., 2014*; *Dimitroglou et al., 2010*; *Mourão et al., 2006*). Many studies reported that feeding MOS to rabbits could increase length, density and improve the organisation of the ileac villi. This suggested a higher rate of intestinal nutrient uptake; thus, improved the growth performance (*Mourão et al., 2006*). The researches which were conducted in chickens, turkeys, pigs, and calves also demonstrated the similar results (*Che et al., 2012a*; *Che et al., 2011*; *Che et al., 2012b*; *Cheled-Shoval et al., 2011*; *Cheled-Shoval et al., 2014*; *Corrigan et al., 2012*; *Ghosh & Mehla, 2012*). Although the identification and the mechanism of actions of MOS remained unclear, existing evidence strongly suggests that MOS may exert biological effects on the intestine regarding tissue compact and increased in structure and activity.

MOS is often prepared by hydrolysis reaction of a mannose-contained glucan polymer, mainly glucomannan and galactomannan (*Cescutti et al., 2002*; *Ganter et al., 1995*; *McCleary, Matheson & Small, 1976*). Glucomannan, a soluble dietary fiber, is a heteropolymer chain of beta-D-glucose and beta-D-mannose which is partially attached with acetyl groups in a molar ratio of 1:1.6 with beta 1–4 linkages. Glucomannan is widely distributed in the tuber or root of the konjac (*Amorphophallus konjac* or *Amorphophallus rivieri*) (*Maeda, Shimahara & Sugiyama, 1980*). Unfortunately, a recent study revealed that the oligosaccharides obtained from the digestion of glucomannan are composed of several random sequences of glucose and mannose residues which hinder the characterization of the oligosaccharides produced (*Cescutti et al., 2002*).

In contrast, galactomannan is an insoluble fiber found in the endosperm of many plant species, such as fruits of coconut trees (*Cocos nucifera*) or coconut meal (as copra meal in dried material), guar gum or guaran from cluster beans (*Cyamopsis tetragonoloba*), and tara gum from Peruvian carobs (*Caesalpinia spinosa*) (*Mathur, 2012*). Another source of galactomannan is yeast cell wall (Saccharomyces cerevisiae), commercially available as Bio-Mos® (Alltech, Inc., Nicholasville, KY, USA). These types of mannan consist of (1-4)-beta-D-mannose repeating units with (1-6)-alpha-D-galactose units attached to the mannose backbone (*Ganter et al., 1995*; *McCleary, Matheson & Small, 1976*; *Sittikijyothin, Torres & Gonçalves, 2005*).

The production of MOS from galactomannan can be performed either by acid or enzymatic hydrolysis. Acid hydrolysis and mechanical hydrolysis are the conventional methods for an industrial scale guar gum production from cluster bean (*Ganter et al., 1995*; *Miyazawa & Funazukuri, 2006*). Although this method was suitable for large-scale production, the sizes of MOS produced cannot be predicted. MOS produced may also require additional purification process to remove the acid and chemical contaminants. Hence, different types of substrate and reaction condition affect the characteristics of MOS product (*Ganter et al., 1995*).

Recently, the production of MOS by enzymatic hydrolysis is of great interested since the product obtained has become more specific and predictable. The enzyme mainly used in the process is beta-mannanase that was found in many organisms including fungus, yeast, and bacteria (*Bourgault & Bewley, 2002*; *Chauhan et al., 2012*; *Cheng et al., 2016*; *Kim et al., 2018*; *Kurakake & Komaki, 2001*; *Larsson et al., 2006*; *Puchart et al., 2004*;

*Rosengren et al., 2014*; *Shi et al., 2011*; *Talbot & Sygusch, 1990*). Beta-mannosidase (E.C. 3.2.1.78) is a hemicellulose-type enzyme that catalyzes random hydrolysis of the internal beta-1,4-glycosidic bond of various mannan-type polymer yielding multiple size beta-1,4-mannooligosaccharides (*Cescutti et al., 2002*; *Ghosh et al., 2015*; *Larsson et al., 2006*; *McCleary, 1988*; *Talbot & Sygusch, 1990*). Several studies have reported that enzymatic production could yield MOS with the degree of polymerisation from 2-6, which is related to those bioactive oligosaccharides that mostly have a low molecular weight or has a low to moderate degree of polymerisation (*Mattaveewong et al., 2016*; *Muanprasat & Chatsudthipong, 2017*; *Muanprasat et al., 2015*; *Yousef et al., 2012*). However, some enzymes produce a low amount of moderate size of MOS (DP 5 and above) from galactomannan substrate compared to those small sizes MOS (DP 2-4) (*Ghosh et al., 2015*; *Rungrassamee et al., 2014*).

*Bacillus subtilis*, a gram-positive bacterium, is well-known for its capability of secreting several beneficial enzymes such as beta-mannanase, xylanase, and glucanase. This bacterium can be isolated from soil, water, and decomposing plant matter. It has also been found in the gastrointestinal tract of animals (*Hong et al., 2009*; *Lefevre et al., 2017*; *Tam et al., 2006*; *Wang & Fung, 1996*). *Bacillus subtilis* is also used in a traditional fermented soybean food (e.g., Natto) in Japan (*Lefevre et al., 2017*; *Wang & Fung, 1996*).

In this study, we reported the production of MOS from pretreated copra meal by enzymatic hydrolysis using recombinant endo-1,4-beta mannosidase derived from *Bacillus subtilis c* AE24. Optimization of the enzymatic production and purification of MOS was elucidated. The screening of bioactive MOS was performed by measuring the effects of of each purified MOS oligomers treatment on a tight junction assembly of an intestinal epithelial cell line.

## MATERIALS & METHODS

### Substrate preparation

Galactomannan substrate was received as a dry granulated copra meal, a residual waste from coconut milk and oil extraction process. Forty grams of copra meal was ground using a high-power blender (Moulinex, France). A finely ground copra meal powder was then subjected to solvent extraction to remove the remaining oil with 200 milliliters of n-hexane, and with sonication for 10 min. The suspension was filtered with a cellulose filter paper (Whatman no.1, Sigma Aldrich, USA.). This process was repeated three times before the copra meal was left to dry in an oven at 60 °C overnight. The dried substrate was resuspended in 400 milliliters of deionized water and then autoclaved at 110 °C and 5 p for 20 min. The product of this step is labelled as G-GalMan.

Thereafter, G-GalMan was sonicated using a probe-type sonicator at 40% power, 2 seconds/1 s pulse, for 300 min. The sonicated products were spun down at $10,000 \times$ g and washed with deionized water twice before resuspending in 100 milliliters of 0.05 M citrate buffer pH 6.5. The final pretreated product is labelled as S-GalMan. Wet weight and dry weight of S-GalMan were measured.

## Structural analysis of pretreatment galactomannan substrate

All the deoiled substrates, including G-GalMan, S-Galman, and digested products were prepared as previously described. All the substrates were subjected to critical point dryer (CPD) and were coated with gold particles prior to the scanning electron microscopy (SEM-EDS) (Jeol JSM-6400 scanning electron microscope; Jeol Ltd., Tokyo, Japan).

## Isolation of *Bacillus subtilis* cAE24

A baiting technique, burying dry copra meal in the soil at different suitable locations, was performed for the isolation of mannanase-produced bacteria. The bacteria were dispersed from the buried copra meal with 3 mL of sterile distilled water, then serially diluted and screened on minimal medium plates containing 1% glucomannan as a carbon source. The positive colony, with clear zone, were then picked and used for enzyme production in minimal medium containing 1% glucomannan, 0.03% magnesium sulfate, 0.1% ammonium sulfate, 0.6%potassium dihydrogenphosphate, and 1% potassium hydrogenphosphate. The crude enzyme produced in the supernatant was collected. Mannanase activity was assayed using Dinitrosalicylic (DNS) colorimetric assay with 0.8% glucomannan as a substrate (*Miller, 1959*; *Vanaja & Shobha Rani, 2007*). The colony with the highest mannanase activity was collected. It was identified by 16s rRNA sequence comparison method.

## Cloning and construction of recombinant mannanase

The amplification of the target mannanase gene was accomplished by using information from the genome database il433616933:611507-613595 of *Bacillus subtilis* strain BEST7613. The polymerase chain reaction was performed using 5′-GGGGAGTTGCATATGTTTAAGA-3′and 5′-GCGGAACGTCTGATTAGAGC-3′ as a forward primer and reverse primer, respectively. The product encoding the mannanase gene was sequenced and submitted to NCBI. DNA sequencing data is available at NCBI via GenBank accession number KY951415. The PCR product was recombined to pGEM-t-easy vector and then transformed into TOP10 *E. coli* using electroporation technique. The recombination plasmid containing the mannanase gene was digested with *XhoI* and *NdeI* restriction endonuclease, and the mannanase gene was subcloned into pET21b expression vector. The final expression plasmid, pRM24, was transformed into *E. coli* BL21 (DE3) expression host (Novagen, Madison, WI, USA). This pRM24 containing *E. coli* BL21 (DE3) was named RM24.

## Purification and Biochemical characterization of RMase24

The crude enzyme production was done with RM24 cultured in $1\times$ LB at 37 °C with 1.0 mM IPTG. Mannanase, RMase24, was collected as extracellular enzyme and concentrated before dialyzing against 20 mM tris-HCl buffer pH 7.5 at 4 °C and then purified through DEAE toyopearl column. The fractions with enzymatic activity were pooled together. Ammonium sulfate was added to give the final concentration of 1 then it was applied onto phenyl toyopearl column. A linear salt gradient from 1.0–0 M ammonium sulfate in 20 mM Tris-HCl pH 7.5 was applied. Fractions that possessed RMase24 activity were collected and pooled. The purity of the enzyme was determined by SDS-PAGE. The activity

of RMase24 was assayed by measuring the total reducing sugar using DNS colourimetric method (*Miller, 1959*). The activity unit was defined as the amount of enzyme that is required to release 1 micromole of reducing sugar from 0.5%w/v glucomannan substrate per minute.

The optimum pH and temperature of purified RMase24 were also explored using 0.4%w/v glucomannan. The optimal pH for RMase24 was measured in a pH range of 2.5–10.0 at 70 °C in 50 mM glycine-HCl buffer (pH 2.5–3.5), citrate buffer (pH 3.5–6.0), phosphate buffer (pH 6.0–8.0) or glycine-NaOH (8.0–10.0). The optimum temperature for RMase24 was determined by assaying enzymatic activity in 50 mM citrate buffer pH 6.0 in a temperature range of 4–80 °C.

For pH stability analysis, the residual enzymatic activity was measured by preincubating the enzyme at 4 °C for 24 h at various pH described above. The thermostability of RMase24 was determined by measuring the residual activity of the enzyme after incubation in 50 mM citrate buffer pH 6.0 at 70 °C from 0 to 6 h. The experiment was performed triplicate. Each data point is the mean, while the error bar is the standard deviation.

## Effects of metal ions and some chemical reagent on the activity of RMase24

The effect of metal ions and some chemical reagent on the activity of RMase24 was determined by measuring the mannanase activity in a reaction containing 0.5% glucomannan, 50 mM citrate buffer (pH 6.0) and 5 mM of some ions including $Na^+$, $K^+$, $Mg^{2+}$, $Co^{2+}$, $Ca^{2+}$, $Mn^{2+}$, $Cu^{2+}$, $Zn^{2+}$, $Fe^{3+}$, and EDTA) or 0.1%(w/v) of surfactants including SDS and TritonX-100 at 70 °C. The activity was measured using DNS assay using glucose as a standard (*Miller, 1959*).

## Production of MOS from a crude RMase24

The digestion of S-GalMan and G-GalMan with crude RMase24 was performed. One hundred micrograms of each substrate were used in a process with various enzyme/substrate concentration ranging from 1unit to 100 units per 1 gram of substrate. After the reactions were incubated for 24 h at 37 °C, digestion mixtures were centrifuged at 10,000 g to remove the undigested debris. The hydrolysate was filtered using filter paper (Whatman cellulose filter paper grade 4; Sigma Aldrich, Carlsbad, CA, USA) and 0.45 micrometers cellulose acetate membrane filter (Whatman cellulose acetate membrane; Sigma Aldrich, Carlsbad, CA,USA). Crude MOS produced from each substrate was collected then lyophilized and measured the dry-weight. The percentage yield was calculated to compare the efficiency between each type of substrate pretreatment.

The soluble MOS product was purified using Biogel-P2 size exclusion column chromatography. The collected MOS fractions were then lyophilized and analyzed using thin layer chromatography (TLC) using acetic: butanol: water solvent system of 3: 3: 2 as mobile phase at 35–40 °C. The separated MOS was visualized by orcinal-sulfuric acid staining solution. The molecular mass determination of tetramer to heptamer MOS was performed using MALDI-TOF mass spectrometer (Solari X, FT mass spectrometry; Bruker, Billerica, MA, USA).

## Determination and identification of bioactive MOS

Each purified MOS ranging from DP4 to DP7 (M4-M7) was filter-sterilized through 0.2 micrometer Whatman uniflo syringe filter (GE Healthcare, Chicago, IL, USA) before being made up to 10 micromolar solutions with Dulbecco's Modified Eagle's Medium (DMEM, Corning life science, USA). The Integrity of the epithelial tight junction was determined by the transepithelial electrical resistance method (TEER) method, with and without $Ca^{2+}$ switch assay. The measurement protocol was described in detail in our previous study (*Muanprasat et al., 2015*). The cell line used in this experiment was T84, a lung metastasized-human colonic adenocarcinoma cell. The cell was growth in DMEM with 10% fetal bovine serum in 12 mm. Transwell (Corning, Corning, NY, USA) culture plates and measured the TEER daily. Once the electrical resistant of the cells became steady, each prepared MOS was treated to the cells. The change in TEER was monitored at 24 h post-treatment. The tight junction reassembling assay was performed with $Ca^{2+}$ switch assay. T84 cells were cultured in DMEM medium until the population of the cells reached 85% confluency on transwell plate, then DMEM medium was substituted with SMEM to remove $Ca^{2+}$ ions. After 24-hour incubation, SMEM was then substituted with prepared DMEM-M5. TEER at different time intervals were measured to monitor the reassembling of the tight junction for 12 h.

## Statistical analysis

Statistical analysis was performed using GraphPad prism7 (GraphPad Software Inc. La Jolla, CA, USA).

# RESULTS

## Isolation of *Bacillus subtilis c*AE24 and cloning of RMase24 gene

We have successfully isolated bacterial strain with high mannanase activity. It was identified by 16s rRNA sequence comparison, which showed a 99% sequence identity with *Bacillus subtillis* strain W_6 (accession number JX462604.1). The isolate was named *Bacillus subtilis* cAE24. Sequemce analysis result of cAE24 and 16s rRNA sequence comparison are available in (Data S1 and Fig. S1). The gene sequence of RMase24 was 1,090 bp in size, encoding 362 amino acid residues (Data S2). The theoretical molecular weight and pI is 40,918.08 Da and 5.80, respectively. The amino acid sequence of RMase24 was blasted with NCBI database and aligned with mannanase sequences from various *Bacillus subtilis* strain. The alignment result showed 99% identity. Three-dimensional structure of RMase24 was constructed using Swiss-PdbViewer, DeepView version 4.1, compared with mannan endo-1,4-beta-mannosidase from *Bacillus subtilis* BEST7613 (accession number BAM49507.1). The sequence alignment and the 3D structure are available in (Figs. S2 and S3).

## Production and characterization of RMase24

A recombinant mannanase gene from *Bacillus subtilis c* AE24 was successfully cloned and expressed. The recombinant enzyme was named RMase24. The crude enzyme was purified through DEAE Toyopearl DEAE650m and Toyopearl phenyl 650 m column chromatography, respectively. The specific activity of RMase24 increased a 3.1 fold, 1,800

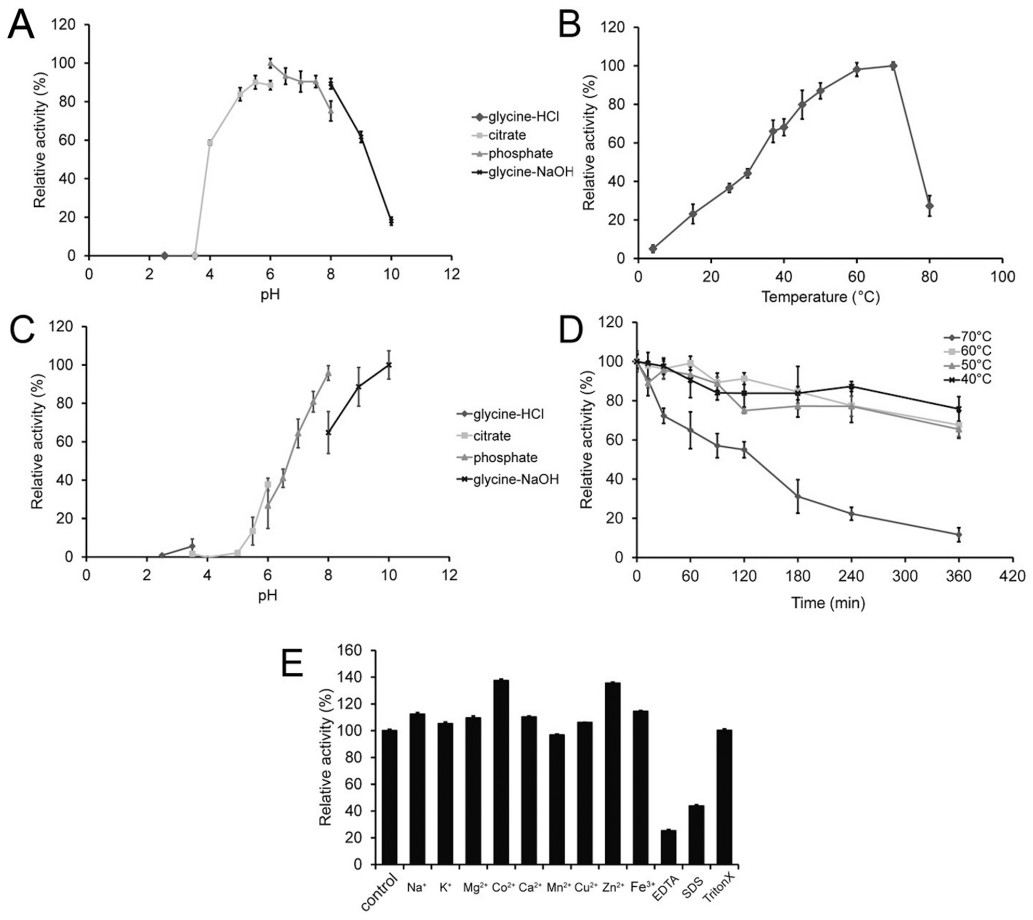

**Figure 1  Optimization, characteristic of purified RMase24.** (A) and (B) reveal the optimum pH and temperature of purified RMase24, respectively. (C) represents the pH stability of RMase24 under a storage at 4C for 24 h. (D) represents thermal stability of RMase24 under a certain temperature for a period of time.

units per milligram protein. RMase24 possesses the ability to work under a wide pH range. RMase24 had over 80% of its maximum activity at pH ranging from 5.0 to 8.0 with the optimum activity in citrate buffer pH 6.0 (Fig. 1A). RMase24 has over 80% activity from 45 to 70 °C, with the optimum temperature at 70 °C. (Fig. 1B). Thus, the broad spectrum of pH and temperature of RMase24 made it possible to set up the reaction under various desirable reaction conditions. Interestingly, purified RMase24 was stable, retaining most of its activity, when stored under alkali condition. The stability of RMase24 significantly dropped when the pH of the storage buffer became more acidic (Fig. 1C).

RMase24 presented stability on a wide range of temperature. The enzyme retained over 75% at temperatures ranging from 40 to 60 °C up to 360 min; however, RMase24 lose 50% off its activity after 120 min at 70 °C and dramatically lose its activity down to 10% at 360 min (Fig. 1D).

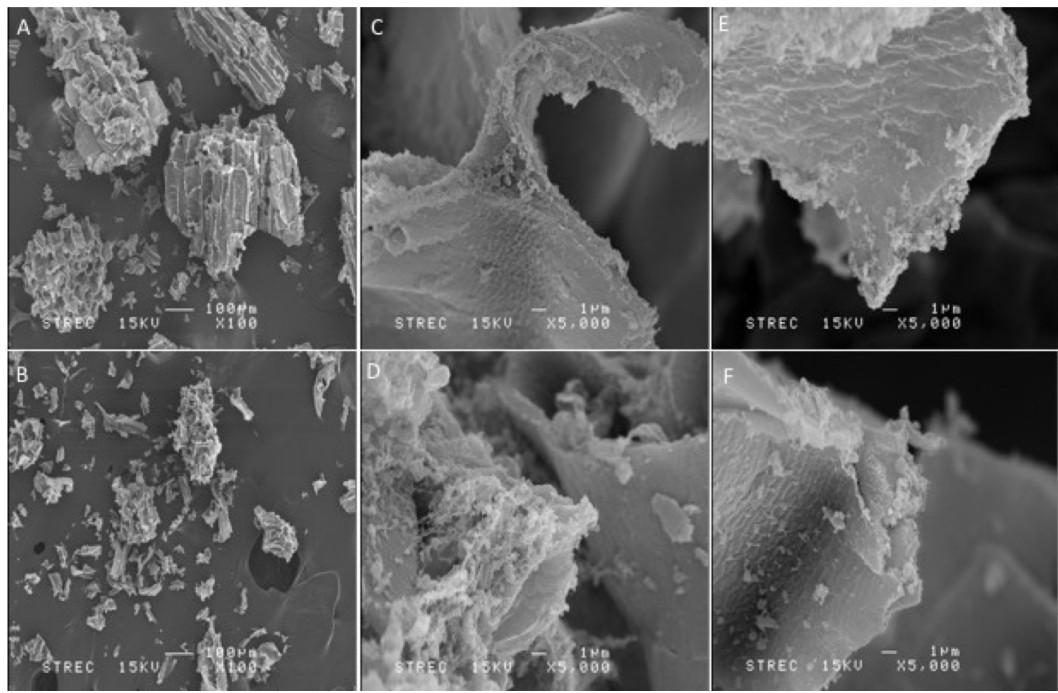

**Figure 2** **SEM analysis of S-GalMan and G-GalMan.** (A) and (B) represent the average fragment size of coconut meal of G-GalMan and S-GalMan, respectively at ×100 magnification. (C) and (D) show the characteristic of the fragment edge and surface of G-GalMan and S-GalMan, respectively. (E) and (F) show the edge and surface structure of G-GalMan and S-GalMan, respectively, after a digestion.

The effects of metal ions also were investigated. The ions that show the highest enhancing effects to RMase24 were cobalt and zinc ions, which increased its activity to about 136 and 137 percent, respectively. Moreover, the results showed that surfactants and chelating agents also affected the activity of RMase24. The reaction contained SDS or EDTA decreases the activity of RMase24 to 25% and 44%, respectively, while Triton X-100 had no effects on RMase24 activity (Fig. 1E).

## Structural analysis of pretreatment galactomannan substrates

The surface structure of each substrate was analyzed by scanning electron microscopy. The result showed different characteristics of fragment surface, edges and size distribution. Sonication pretreatment gave smaller average fragment size when compared with the untreated substrate (Figs. 2A–2B). At higher magnification, different characters of fragment surface and edges from each type of substrate was observed (Figs. 2C–2D). S-GalMan had more irregular amorphous shape edges, and there was a considerable amount of amorphous structure on the surface of the substrate, while G-GalMan has less of these characteristics. Interestingly, after the digestion of S-GalMan with RMase24, these small fragments were reduced, and the edge and surface of the fragments became smooth. In contrast, digestion of G-GalMan did not change the overall structure observed by SEM much (Figs. 2E–2F).

## MOS production and purification, the effect of substrate pretreatment on the production, and the structural prediction of M5

The percentages yield of total MOS produced from S-GalMan and G-GalMan were 8.25% and 5.09% (dry basis), respectively. Thin layer chromatography analysis of digested products showed a spectrum of MOS with size ranging from DP2 to DP6 and MOS with a size larger than DP6, comparing to standard MOS ladder. At a concentration of 10 units of RMase24 per 1 gram of substrate, the production of MOS from S-GalMan gave a higher yield of MOS when compared to G-GalMan at every incubation period. S-GalMan could be digested more readily than G-GalMan, liberating a higher amount of MOS, within the first 6 h (Fig. 3A).

Determination of appropriate RMase24 units required for optimum MOS production was then performed with S-GalMan to determine the optimal enzyme/substrate ratio. The results showed that the enzyme/substrate ratio used in the digestion affected the pattern of MOSs produced (Fig. 3C). The enzyme-substrate ratio of 10 units per 1 gram of dried substrate showed the highest yield of MOS DP4 to DP6. Interestingly, at higher concentration of RMase24 we observed a reduction of DP6 and DP9 (Fig. 3B).

## M5 affect the tight junction integration of an epithelial tissue

MOS DP4 (M4) to DP7 (M7) was successfully purified through Biogel-P2 column (Fig. 4A) and their molecular mass confirmed by mass spectrometry. The arrow in Figs. 4C–4F indicated a peak of molecular mass of each MOS with sodium ion; M4-Na$^+$ = 679.123 m/z, M5-Na$^+$ = 851.2646 m/z, M6-Na$^+$ = 1013.3171 m/z, M7-Na$^+$ = 1175.3700 m/z, respectively. Tight junction integration was measured on T84 cells treated with M4 to M7 using TEER assay. The result demonstrated that treating the cells with M5 significantly increase the relative percentage of TEER ($p < 0.0001$). This demonstrated that M5 can enhance tight junction integration of T84 cell while treating cells with M4, M6 and M7 did not demonstrate any significant effect on TEER comparing with vehicle (Fig. 5A). The effects of M5 over cellular tight junction was distinguished from the cellular proliferation by calcium depletion experiment where calcium ions were removed from the culture medium once the growth of the cell reached 85% confluency, replacing DMEM with SMEM. After 24 h of incubation in SMEM, the culture medium was replaced with DMEM medium containing calcium ions and M5. TEER was monitored every 15 min for 12 h. The results revealed that M5 could significantly increase the rate of tight junction reassembly of T84 cells when comparing to vehicle (control, non-treatment) group, without M5 (Fig. 5B).

## DISCUSSION

In this study, recombinant mannanase, RMase24 together with a proper substrate pretreatment and enzyme-substrate ratio showed a better production yield of moderate MOSs compared with previous studies (*Ghosh et al., 2015*; *Rungrassamee et al., 2014*). These results may result from the higher exposed surface area arose from the additional sonication step that is accessible to enzymatic hydrolysis. Interestingly, the enzyme-substrate ratio affected the yield and size distribution of MOS products. We observed a reduction in the band intensity of M4, M6, and M9, while Intensity of other bands including M2, M3, and

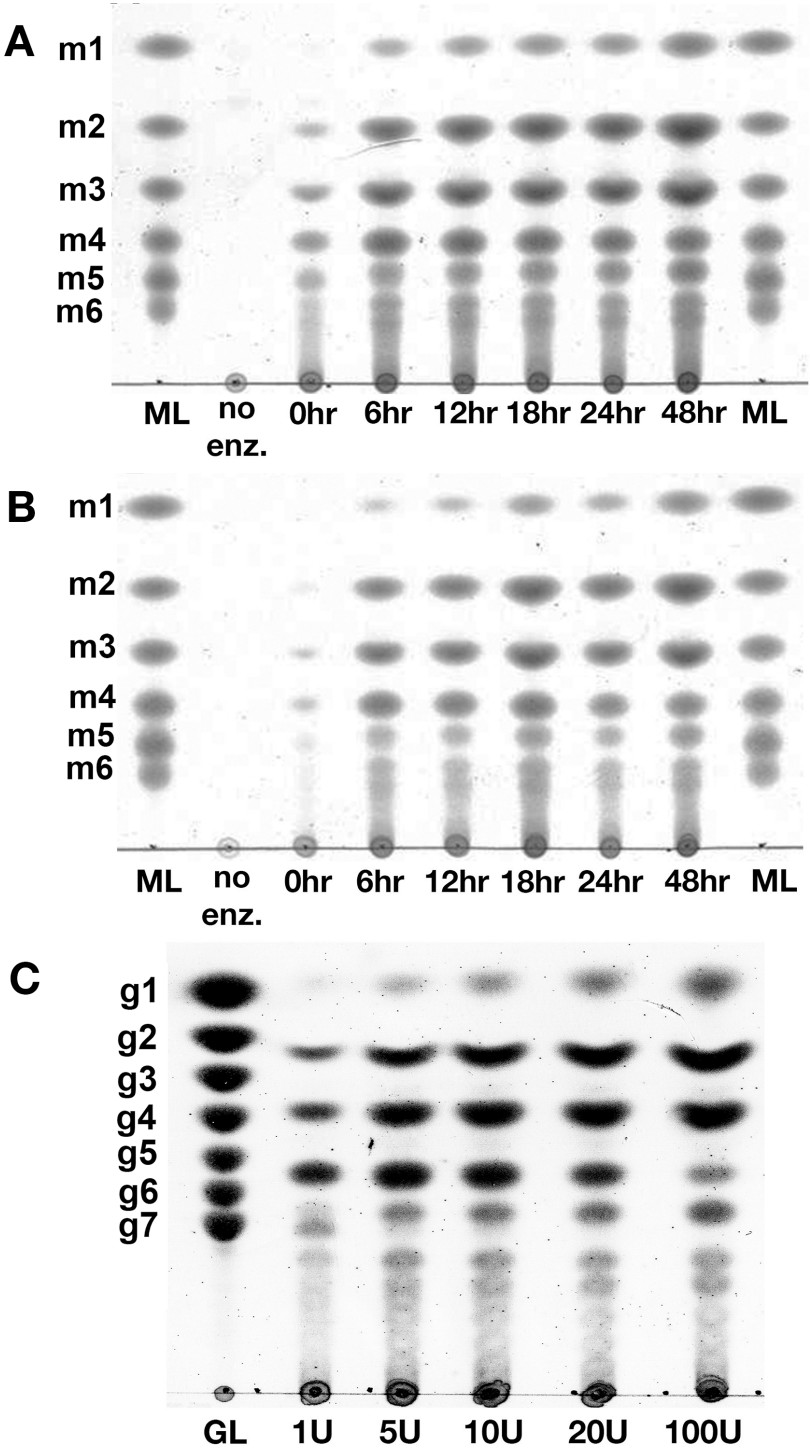

**Figure 3** **Product spectrum at different incubation time and enzyme substrate ratio.** A comparison between a digestion of S-GalMan (A) and G-GalMan (B) with 10 units per 1 gram of substrate. (ML; Mannan oligosaccharides ladder, No; no RMase24 in the reaction, m1-m6 represents mannose to mannohexose, respectively) (C) The pattern of MOS production at different RMase24 and substrate ratio, the substrate was fixed at 1 gram. (GL; Glucooligosaccharides ladder, g1-g7 represent glucose to glucoheptose, respectively).

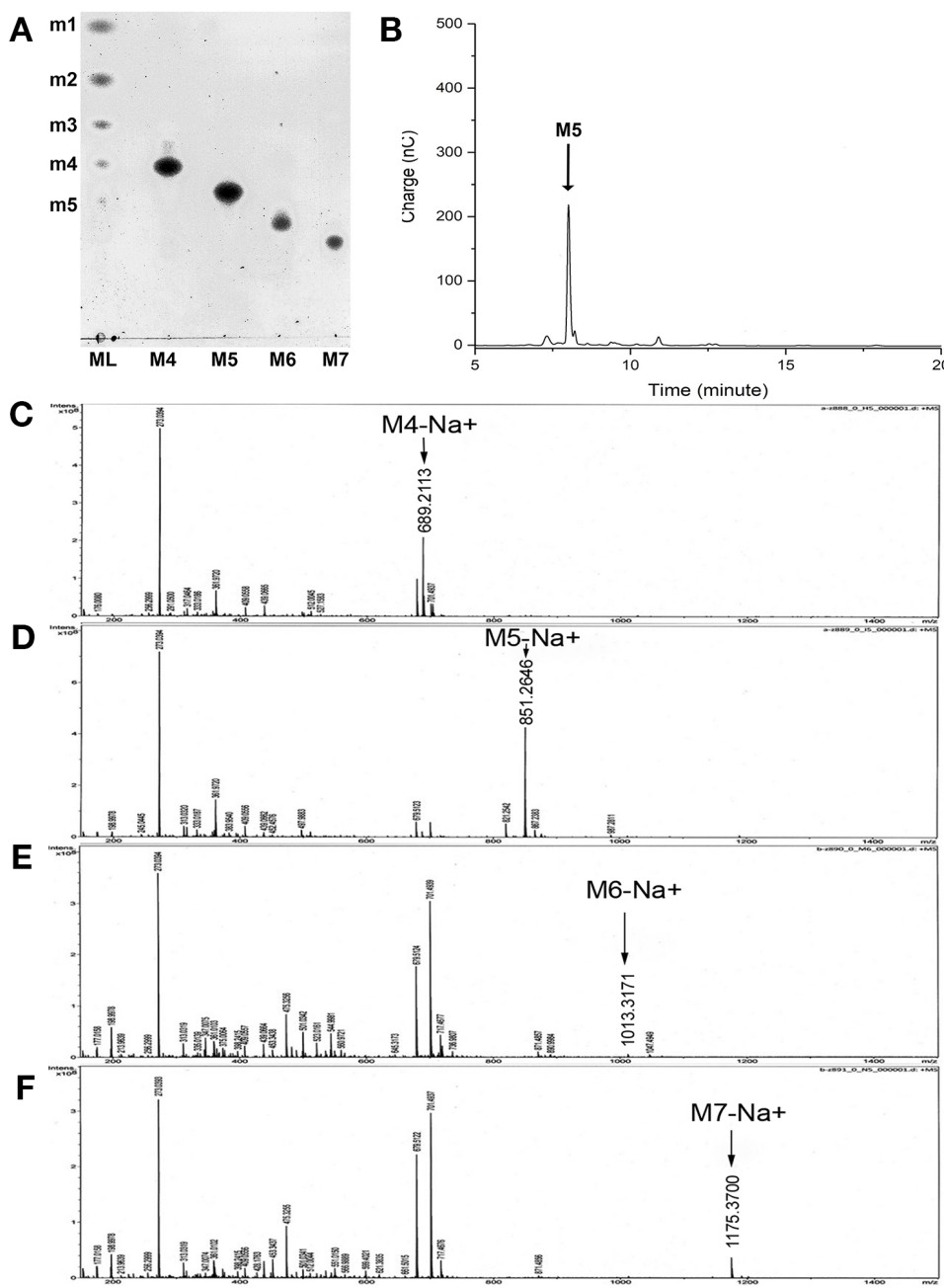

**Figure 4 Purified MOS, from M4 to M7.** (A) Purified MOS, from M4 to M7 analyzed on TLC. ML stands for Mannooligosaccharides ladder, m1–m5 represent standard mannose to mannopentose, respectively. (B) High-performance anion exchange chromatography (HPAEC-PAD) chromatogram of purified M5. The black arrow indicates the peak of M5. (C–F) High-resolution mass spectrometry of purified M4, M5, M6, and M7, respectively. The arrow indicated a peak of molecular mass of each MOS with sodium ion; M4-Na = 679.123 m/z, M5-Na = 851.2646 m/z, M6-Na = 1013.3171 m/z, M7-Na = 1175.3700 m/z.

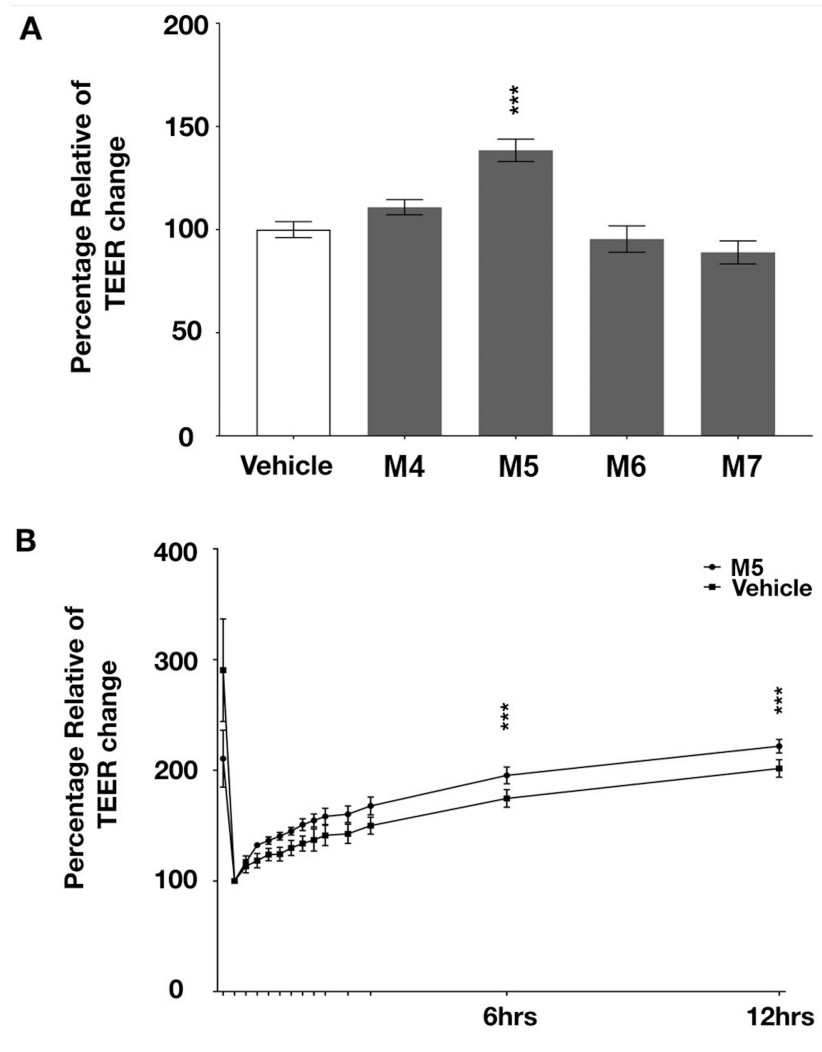

**Figure 5  Effects of MOS on tight junsssction integration of T84 cells.** (A) Transepithelial electrical resistant (TEER) result of T84 cells treated with 10 micromolar of purified M4 to M7, compared to vehicle (non-treatment group). M5 significantly increased TEER of T84 cells (one-way ANOVA, $p < 0.00001$, $n = 3$). (B) Determination of tight junction reassembly of T84 affected by a treatment of 10 micromolar M5 compared to vehicle using $Ca^{2+}$ swtich assay. M5 significantly recovered TEER of T84 cells after the destruction of the tight junction. (two-way ANOVA, $p < 0.00001$, $n = 4$ s).

M5 increased as the enzyme over substrate ratio increases, Fig. 3C. At high enzyme over substrate ratio, the availability free enzyme in the solution may cause further hydrolysis of some MOS products that remains a good substrate for the enzyme. These results indicate that the substrate specificity of RMase24 is different for each MOS species being produced.

Biological activity of MOS have been previously reported (*Che et al., 2012a*; *Che et al., 2011*; *Che et al., 2012b*; *Cheled-Shoval et al., 2011*; *Cheled-Shoval et al., 2014*; *Corrigan et al., 2012*; *Dimitroglou et al., 2010*; *Genc et al., 2007*; *Ghosh et al., 2015*; *Ghosh & Mehla, 2012*; *Grisdale-Helland, Helland & Gatlin III, 2008*; *Mansour et al., 2012*; *Mourão et al., 2006*;

*Staykov et al., 2007*). However, these studies used a mixture of MOS of different sizes. Thus, the biologically active MOS species were not identified. In this study, we separated and purified each MOS species prior to biological activity analysis. We discovered that M5 was the only molecule that significantly enhanced the tight junction integration of T84 cells. This effect was distinguished from the effect of cell proliferation by $Ca^{2+}$ switch assay (Fig. 5B). This finding indicated that M5 was playing the vital role in increasing the tight junction of epithelial tissue. Hereafter, further investigation is required to identify the structure and mechanism of action of M5.

## CONCLUSIONS

Production of moderate size of MOS, DP4-DP7, by endo-mannanase can be enhanced using proper substrate pre-treatment. The enzyme-substrate ratio used for MOS production is a crucial factor affecting yield and size distribution of MOS. Specific MOS species contains biological activity. The structure of action of the biologically active MOS species should be determined and mechanism of action further investigated.

### Funding

This work was supported by the 100th Anniversary Chulalongkorn University Fund for Doctoral scholarship, the office of Graduate School, Chulalongkorn University. The funders had no role in study design, data collection and analysis, decision to publish, or preparation of the manuscript.

### Grant Disclosures

The following grant information was disclosed by the authors:
100th Anniversary Chulalongkorn University Fund for Doctoral scholarship, the office of Graduate School, Chulalongkorn University.

### Competing Interests

The authors declare there are no competing interests.

### Author Contributions

- Chatchai Nopvichai and Thanapon Charoenwongpaiboon performed the experiments, analyzed the data, prepared figures and/or tables.
- Navaporn Luengluepunya performed the experiments, bacillus subtilis cAE24 screening.
- Kazuo Ito and Rath Pichyangkura conceived and designed the experiments, analyzed the data, contributed reagents/materials/analysis tools, authored or reviewed drafts of the paper, approved the final draft.
- Chatchai Muanprasat conceived and designed the experiments, contributed reagents/materials/analysis tools, authored or reviewed drafts of the paper, approved the final draft.

## DNA Deposition

The following information was supplied regarding the deposition of DNA sequences:

The DNA sequencing data is available at GenBank: KY951415

## Data Availability

The raw data of the TEER experiment in Fig. 5 is available as a Supplemental File.

## Supplemental Information

Supplemental information for this article can be found online at http://dx.doi.org/10.7717/peerj.7206#supplemental-information.

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
