# Peer review of "Production and purification of mannan oligosaccharide with epithelial tight junction enhancing activity"

_PeerJ, doi:10.7717/peerj.7206_

## Round 0.1 · original submission · Major Revisions

· Academic Editor

Major Revisions

I invite you to revise your paper according to the comments made by the reviewers and resubmit your paper.

Reviewer 1 ·

Basic reporting

In this study, a targeted mannanase gene derived from Bacillus subtilis cAe24 isolate was successfully constructed and expressed into BL21 strain. The recombinant enzyme, RMase24, was subjected to not only examine its activities, but the RMase24 was also utilized for producing mannan oligosaccharide (MOS) from a copra meal. Based on the obtained data, it seemed that MOS5 was significantly increased tight junction integration of epithelial cells when compared with negative control. Overall, this is a very interesting manuscript with a comprehensive investigation, therefore it will certainly attract numerous citations in the future. Also, the scientific background, statistic and the research as a whole are generally sound. However, the manuscript should be thoroughly improved in writing, and some requirements should be revised to meet the standard of publication.

Introduction: Since the RMase24 was expressed from a targeted mannanase gene derived from Bacillus subtilis cAe24 in this study, not only general backgrounds of this isolate but also bacterial/microbial mannanases should be described. If there is a publication of this isolate previously, please obviously state in the manuscript.

Materials and Methods: Strains, media, general experimental conditions, chemicals, etc. should be added in a separate section. More performed information should be described for sections of SEM-EDS (line 126-128) and effects of metal ions and some chemical reagent on the activity of RMase24 (line 159-162). Writing style should be in passive (for example, revision should be applied for line 113-115, 117-119, etc.). Specific information should be stated (for example, whether “BL21” in line 142 is E. coli BL21 and which source of this strain? When is “the digestion reactions were completed” in line 166? After 24 hrs?). More performed information on TEER measurement should be supplemented.

Results: To comprehensively understand the unique features of RMase24, it is interesting and necessary to know about Bacillus subtilis cAe24 isolate and in silico information of the targeted mannanase gene derived from this isolated. Therefore, I strongly recommend adding this section in the revised manuscript.

Discussion: The authors not only mostly repeated the description of result section, but discussion and comparison with others are largely lacking. Please discuss more bacterial/recombinant mannanase, MOS production, and purification, substrate resources, etc. Any connection of enzyme activities and its bioactive function may be clarified from this study? It is also interesting to discuss and compare with other publications about the mechanism of action of MOS5 as well as its potential application in the future.

Experimental design

For a logical and comprehensive story, preliminary examinations regarding Bacillus subtilis cAe24 and targeted mannanase gene derived from this isolate, leading to the conduction of this study, should be represented as I mentioned above.

Validity of the findings

There was no data for statements from line 211-215.

Figure 1: Figure legend should provide brief information on a specific method to obtain the data. Figure 1C is confusing and may overlap with Figure 1A? Please clarify obviously the statement for this results in line 220-223. Please add statistical analysis, especially for Figure 1E.

Figure 2: If Figure 2C and D are magnifications of Figure 2A and B, respectively? If so, which parts in Figure 2A and B were magnified?

Figure 3: This figure seemed hard to see “size ranging from M2 to approximately M12” as the statement in line 253 of the text. Also, which data did the authors claim for “The percentage yield of total MOS production from S-GalMan and G-GalMan were 8.25% and 5.09% the substrate (dry basis), respectively” (line 256-257).

Additional comments

There are many typos in wording and grammar throughout the manuscript, therefore it should be revised by a native speaker.

Specific reference should be cited immediately after the statement. For example, “Several reports revealed that MOS supplement could improve growth performance and body weight in various animals” in line 52-53, “MOS is often prepared by hydrolysis reaction of a mannose-contained glucan polymer, mainly glucomannan and galactomannan” in line 68-69, etc.

Citation in the main text should be revised with the surname of reference authors. For example, “Mourao et al. 2006” instead of “Mourao JL 2006” in line 61 and 62, “Maeda & Sugiyama1980” instead of “Masaakira MAEDA 1980” in line 73, etc.

The references should be in an inconsistent format. For example, line 353, 361, 365, 406 (capitalized in journal name); line 355 (capitalized in paper title and journal name); line 377, 400 (abbreviated in journal name); line 385 (species name should be in italics); etc.

Reviewer 2 ·

Basic reporting

In this study, Nopvichai and colleagues produced and tested the MOS. They found MOS could enhance the tight junction integration T84 cells. In general, the authors need to do a better job in organizing and writing the text. To be specific, I have some concerns about this work that should be addressed before further consideration.

1. Previous study (Silvia Torrecillas et al., Fish & Shellfish Immunology, 2013) have shown that MOS could enhance intestinal epithelial barrier health status, including enhancing the integration of tight junction. The authors should discuss and compare previous study with the current one to show the improvement or importance of their work.

2. The authors should be more careful with the references. There are a lot of text missing references. Meanwhile, it is inappropriate to put all the references at the end of one paragraph. The authors should specify the references to each point.

3. Figure 5: The authors should provide following information of this figure. a) The concentration of MOS used in this experiment. b) N (biological repeats) for each group. c) The authors only mentioned the software for statistic analysis. They should provide the statistic method used in the analysis.

4. Since MOS had impact on tight junction, the authors should test the expression level of certain tight junction genes via RT-qPCR and westernblot.

Experimental design

no comments

Validity of the findings

no comments

Additional comments

no comments

---

## Round 0.2 · accepted · Accept

· Academic Editor

Accept

I advise that your paper is now acceptable in its present form for publication in PeerJ.

#